# A stochastic vs deterministic perspective on the timing of cellular events

Lucy Ham[1,2,6], Megan A. Coomer[1,2,6], Kaan Öcal[3,5], Ramon Grima [4] & Michael P. H. Stumpf [1,2] ✉

Cells are the fundamental units of life, and like all life forms, they change over time. Changes in cell state are driven by molecular processes; of these many are initiated when molecule numbers reach and exceed specific thresholds, a characteristic that can be described as "digital cellular logic". Here we show how molecular and cellular noise profoundly influence the time to cross a critical threshold—the first-passage time—and map out scenarios in which stochastic dynamics result in shorter or longer average first-passage times compared to noise-less dynamics. We illustrate the dependence of the mean first-passage time on noise for a set of exemplar models of gene expression, auto-regulatory feedback control, and enzyme-mediated catalysis. Our theory provides intuitive insight into the origin of these effects and underscores two important insights: (i) deterministic predictions for cellular event timing can be highly inaccurate when molecule numbers are within the range known for many cells; (ii) molecular noise can significantly shift mean first-passage times, particularly within auto-regulatory genetic feedback circuits.

Cellular dynamics are not deterministic[1]. Observations of RNA or protein numbers in single cells show that their variation with time is stochastic[2–4]. This noise originates from biomolecular processes that are unaccounted for[5,6], as well as from the inherent random timing of individual molecular events which manifests most clearly in low copy number conditions (intrinsic noise)[7,8]. In spite of this randomness, cells orchestrate their processes with remarkable precision and robustness, likely because of mechanisms that either exploit or suppress intrinsic noise[9,10].

Mathematical modelling based on the Chemical Master Equation (CME), a probabilistic description of chemical reaction kinetics, has led to significant progress in understanding how the stationary and non-stationary distributions of molecule numbers in a single cell vary as a function of the rate parameter values[11–15]. Nevertheless, these studies do not directly address a central question: when will the level of some molecule first cross a critical threshold? It is known that regulatory proteins often need to reach critical threshold levels to trigger downstream processes[16–23]. For example, in cellular development and differentiation, progression through different phases of the cell cycle only occurs once certain cyclin proteins surpass a prescribed threshold[24,25]. Similarly, many physiological processes are triggered by calcium signalling and rely on the concentration of calcium ions to exceed a certain threshold[26–29]. Another example is p53-mediated cell apoptosis which is triggered when p53 proteins reach a critical value; below this value, cells enter the G1 arrest phase[30,31] (Fig. 1A). In the context of RNA it has been shown that microRNAs (miRNAs) can create mRNA threshold levels, below which protein production is suppressed[32,33]. From an experimental perspective threshold crossing can, however, be understood more generally. For example, many events in cells can be timed by the turning on or off of a molecular fluorescent marker; since the observed molecular levels are of a continuous nature, a threshold is necessary to define a discrete event from these markers. Examples of processes that have been timed in this way include entry into competence in bacteria, different phases of lysis by

[1]School of BioSciences, University of Melbourne, Parkville, Australia. [2]School of Mathematics and Statistics, University of Melbourne, Parkville, Australia. [3]School of Informatics, University of Edinburgh, Edinburgh, UK. [4]School of Biological Sciences, University of Edinburgh, Edinburgh, UK. [5]Present address: School of BioSciences, University of Melbourne, Parkville, Australia. [6]These authors contributed equally: Lucy Ham, Megan A. Coomer. ✉e-mail: mstumpf@unimelb.edu.au

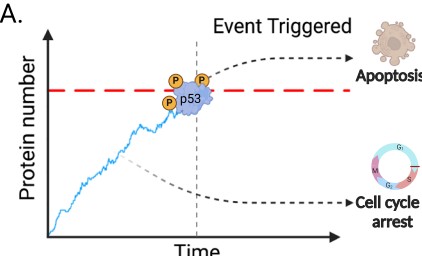

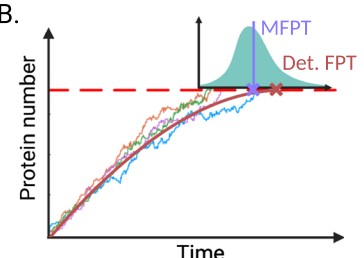

**Fig. 1 | First-passage time processes in cellular biology. A** Many processes rely upon regulatory proteins or molecules attaining critical threshold levels. Once this threshold is surpassed, a biological event is triggered. The time at which this occurs is known as the *first-passage time* (vertical grey dashed line). For example, upon DNA damage, the activation of p53 can elicit two potential cellular responses: cell-cycle arrest or apoptosis. Below a defined threshold of p53 expression, the cell undergoes cell-cycle arrest, whereas apoptosis is induced when the p53 expression surpasses a threshold. (**B**) The underlying process governing the regulatory molecule is stochastic and hence variability in the event timing is expected. This results in a distribution of first-passage times (shown in turquoise), which is conditional on the initial number of protein molecules in the system; here the initial protein number is 0. The mean first-passage time (MFPT; solid purple line and cross) can be compared with the deterministic FPT (red cross) which is the time to reach the target molecular number as estimated by integration of deterministic reaction rate equations (red solid line). (**A**) and (**B**) were created with BioRender.com.

phage, cell cycle phases, and loss or gain of pluripotency in stem cells (for a review see ref. 34).

The time for a molecule number to reach a certain target value is a stochastic variable whose properties can be understood using first-passage time (FPT) theory[35]. Of particular importance is the mean first-passage time, the average time to reach a target value. Analytical expressions for the mean FPT have been derived for one variable models[36,37], and systems with many states connected in a simple way[38–40]. For more complex models, as common in biology, the exact analytical approach is often unfeasible and thus numerical computation, using the Finite State Projection (FSP) method[41,42], or Monte Carlo simulation using the Stochastic Simulation Algorithm (SSA)[43], is typically far more practical. Approximations, such as moment closure approaches, might be feasible too[44,45]. Using one or more of these approaches, several studies have investigated the mean FPT (and higher-order moments and distributions of the FPT) as a function of the rate parameter values[13,46–53]. Despite these advances, biochemical systems have been, and are still, studied by means of deterministic ordinary differential equations due to their simplicity and ease of use[54]. For systems where the counts of molecules are large and reactions are taking place nearly continuously, the dynamics is approximately deterministic[55,56]. However, for systems with small volume and discrete molecule counts, such as the interiors of biological cells, dynamics are stochastic and the validity of deterministic modelling becomes less obvious. Within the deterministic framework, it is also possible to estimate the time to reach a certain molecule threshold by numerical integration of the differential equations. The differences between estimates of FPTs using deterministic and stochastic modelling frameworks have not previously been studied systematically, except in special cases (which we discuss below). This is an important question given the ubiquity of deterministic models of cellular biology and the fact that comparison of the two estimates would provide a direct means to study the role played by intrinsic noise in the timing of cellular events.

There are specific limits where the relationship of the deterministic and stochastic predictions for the mean time to reach a target molecular threshold are well understood. If for a given initial condition, the target molecule number is outside of the range predicted by the deterministic model in finite time then clearly the latter's prediction for the mean time is undefined whereas a stochastic model will typically predict a finite value. An extreme example is given by extinction processes, where initially the number of molecules is larger than zero and the target value is zero, a value that the deterministic model only approaches asymptotically[57,58]. If the target molecule number is within the range predicted by the deterministic model in finite time then, if intrinsic noise is very small (molecule numbers are sufficiently large), the trajectories obtained from the SSA are close to the temporal variation of the mean molecule numbers predicted by the deterministic model[36,56] and hence the stochastic and deterministic approaches will necessarily predict the same mean time to reach a threshold molecule number.

The relationship between deterministic and stochastic predictions of the mean time to reach a threshold value is, however, unclear when the molecule numbers are small—the typical case inside cells. Even if the mean molecule numbers of the deterministic and stochastic models are the same for all times (as for example is the case for systems with linear propensities[37]), when intrinsic noise is large, the trajectories of the SSA will not closely follow the deterministic prediction. For systems with nonlinear propensities (such as those with bimolecular reactions) the differences are likely even more pronounced because the mean molecule number predictions of the two models disagree, which will be reflected in the trajectories of the SSA[59]. Hence, generally, it is difficult to say if the mean time for trajectories of the SSA to reach a designated target level (as measured by the mean FPT) is the same, larger or smaller than the time predicted by the deterministic model. An illustration comparing the two model predictions is shown in Fig. 1B.

We shed light on this problem by computing the difference in trigger time predictions of deterministic and stochastic models of a variety of biochemical processes with the constraint that the target molecular value is larger than zero and smaller than the steady-state mean molecular number predicted by deterministic models. The analysis enhances our understanding of the role played by intrinsic noise in cellular event timing.

## Results
### Mathematical framework
We can formulate the question of interest as a FPT problem: for a continuous-time Markov process (CTMP) $\mathbf{x}(t)$, we are interested in the time it takes $\mathbf{x}(t)$ to first arrive at some subset $\mathbf{Y}$ of the discrete state-space, given the system started from $\mathbf{n} \notin \mathbf{Y}$. In other words, we are interested in first-passage times,

$$\tau_{\mathbf{n}} = \inf\{t \geq 0 : \mathbf{x}(t) \in \mathbf{Y} \,|\, \mathbf{x}(0) = \mathbf{n}\}. \tag{1}$$

**Moments of the FPT distribution.** We derive moments of the FPT distribution for any CTMP $\mathbf{x}(t)$, providing a fresh intuitive proof of a result that has been presented in the literature[36,49]. Fix an initial state $\mathbf{n} \notin \mathbf{Y}$ and let $\tau_{\mathbf{nz}}$ be the waiting time to reach the absorbing state $\mathbf{Y}$ from $\mathbf{n}$ given that $\mathbf{z}$ is the next state visited and $\mathbf{z} \notin \mathbf{Y}$. The random variable $\tau_{\mathbf{nz}}$ can be decomposed into the sum of two independent

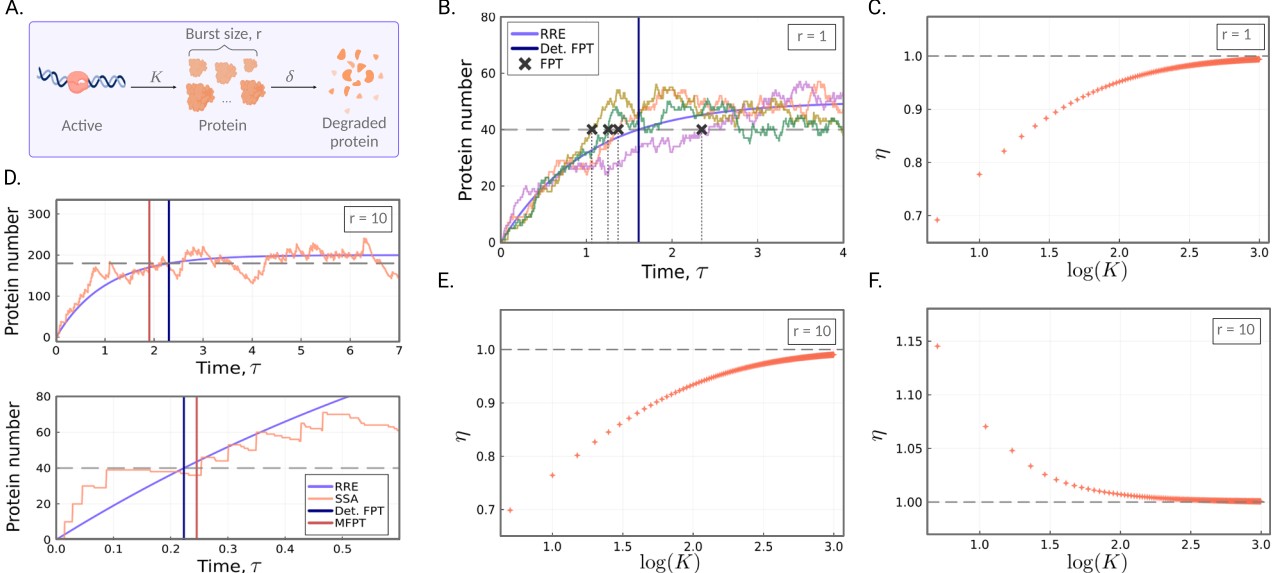

**Fig. 2 | Stochastic vs. deterministic waiting times to reach a target protein number for the bursty birth–death process. A** An illustration of the bursty birth–death process (as given by the reaction scheme in Eq. (8) with burst size $r \geq 1$). The schematic was created with BioRender.com. **B** Example trajectories of the simple birth–death process (multi-coloured) as simulated by the SSA. The FPTs of the trajectories are shown as the dark grey crosses, while the deterministic FPT is shown as the blue vertical line. Here $r = 1$, $\rho = 0.8$, $K = 50$ and $\delta = 1$, so that the target $N = 40$ (grey dashed line). **C** The ratio $\eta = \langle \tau_0 \rangle / \tau_d$ of the stochastic MFPT, $\langle \tau_0 \rangle$, to the deterministic FPT, $\tau_d$, as a function of increasing $K$. The stochastic mean waiting time is computed according to Eq. (9), and the deterministic waiting time is computed according to Eq. (11). Model parameters are $\rho = 0.8$, $K$ varies from 2 to $10^3$, and $\delta = 1$. **D** (top) Waiting times when target $N$ is equal to a large proportion (here

$\rho = 0.9$) of the steady-state mean. A representative time series of the bursty birth–death process is shown in orange, and the solution of the reaction rate equation in purple. Here $r = 10$, $K = 20$ and $\delta = 1$, so that the target $N = 180$ (shown as the grey dashed line). The stochastic MFPT (red vertical line) is smaller the deterministic FPT (blue). (bottom) Parameters are the same as above except now $\rho = 0.2$ and the target is $N = 40$. **E** The ratio $\eta = \langle \tau_0 \rangle / \tau_d$ of the stochastic MFPT, $\langle \tau_0 \rangle$, to the deterministic FPT, $\tau_d$, as a function of increasing $K$, for a high threshold value. The number of initial proteins is 0, $\rho = 0.9$, $r = 10$, $K$ varies from 2 to $10^3$, and $\delta = 1$. **F** Same as (**E**), except now $\rho = 0.2$. Stochastic waiting times are computed according to our FSP approach (refer to the Methods section below), and deterministic waiting times are computed analytically.

random variables,

$$\tau_{\mathbf{nz}} = d_{\mathbf{n}} + \tau_{\mathbf{z}}, \qquad (2)$$

where $d_{\mathbf{n}}$ is the time until the first jump (the exponentially distributed dwelling time) and $\tau_{\mathbf{z}}$ is the waiting time to reach the absorbing set $\mathbf{Y}$ from the state $\mathbf{z}$. Let $\mathbf{A}$ denote the state transition matrix for $\mathbf{x}(t)$ (Supplementary Information, Section 1.1). The expected time until the next jump is $-1/A_{\mathbf{nn}}$, and the probability of transitioning to state $\mathbf{z}$ equals $-A_{\mathbf{zn}}/A_{\mathbf{nn}}$, where here $A_{ij}$ is the $ij$th element of $\mathbf{A}$. For any $k \geq 1$, we can compute the $k$th raw moment of the waiting time $\tau_{\mathbf{n}}$ to reach $\mathbf{Y}$ from $\mathbf{n}$ as follows. By the Law of Total Expectation, we have that

$$\mathbb{E}[\tau_{\mathbf{n}}^k] = -\sum_{\mathbf{z} \neq \mathbf{n}} \mathbb{E}[\tau_{\mathbf{nz}}^k] \frac{A_{\mathbf{zn}}}{A_{\mathbf{nn}}}. \qquad (3)$$

Using Eq. (2), the Binomial Theorem, and the independence of $d_{\mathbf{n}}$ from $\tau_{\mathbf{z}}$, it follows that,

$$\mathbb{E}[\tau_{\mathbf{n}}^k] = -\sum_{i=0}^{k-1} \binom{k}{i} \frac{(k-i)!}{(-A_{\mathbf{nn}})^{k-i}} \sum_{\mathbf{z} \neq \mathbf{n}} \mathbb{E}[\tau_{\mathbf{z}}^i] \frac{A_{\mathbf{zn}}}{A_{\mathbf{nn}}} - \sum_{\mathbf{z} \neq \mathbf{n}} \mathbb{E}[\tau_{\mathbf{z}}^k] \frac{A_{\mathbf{zn}}}{A_{\mathbf{nn}}}, \qquad (4)$$

for $\mathbf{n} \notin \mathbf{Y}$, where here we have also used the fact that $d_n$ is exponentially distributed with rate $-A_{\mathbf{nn}}$. Now moving the rightmost sum to the left and multiplying both sides of Eq. (4) through by $A_{\mathbf{nn}}$, it follows that

$$\sum_{\mathbf{z}} \mathbb{E}[\tau_{\mathbf{z}}^k] A_{\mathbf{zn}} = -k \mathbb{E}[\tau_{\mathbf{n}}^{k-1}], \qquad (5)$$

where here we have used Eq. (3) from right to left with $k-1$ in place of $k$. Recalling that $\tau_{\mathbf{z}} \equiv 0$ for $\mathbf{z} \in \mathbf{Y}$ and that $\mathbf{n} \notin \mathbf{Y}$, we can write Eq. (5) for

all initial states $\mathbf{z} \notin \mathbf{Y}$ concisely as

$$\mathbf{A}_{\mathbf{Y}}^{T} \mathbb{E}[\boldsymbol{\tau}^k] = -k \mathbb{E}[\boldsymbol{\tau}^{k-1}] \quad \text{with} \quad \mathbb{E}[\tau^0] = 1, \qquad (6)$$

where $\mathbf{A}_{\mathbf{Y}}^{T}$ denotes the matrix $\mathbf{A}^{T}$ with rows and columns corresponding to $\mathbf{Y}$ removed. Solving this (matrix) recurrence relation gives,

$$\left( \mathbf{A}_{\mathbf{Y}}^{T} \right)^k \mathbb{E}[\boldsymbol{\tau}^k] = k!(-1)^k \mathbf{1}, \qquad (7)$$

which agrees with the result that can be obtained from the Backward Chemical Master Equation (see Supplementary Note 1). We use a modification of the FSP algorithm to compute mean FPTs (MFPTs) via Eq. (7) (see the "Methods" section below). The approach also extends to the full FPT distribution over a set of initial states (see Supplementary Note 1). Our approach allows us to simultaneously compute FPT statistics, for *all* initial molecule numbers $\mathbf{n} \notin \mathbf{Y}$, significantly reducing computation time in comparison to Monte Carlo simulations; refer to Supplementary Note 5, where we compare CPU times for three models of biological relevance, including enzyme kinetic examples and a compartmental model of disease spread. We remark that the Forward CME can be used to compute FPT distributions by way of a modified FSP, however, requires recomputation for each initial state[42]. Note that Eq. (7) can also be solved analytically in some cases (Supplementary Notes 2 and 3).

## Stochastic vs deterministic calculations

Understanding the average behaviour of a stochastic system is crucial for predictive modelling and system characterisation. We focus here on the relationship between deterministic and stochastic predictions of the average time to reach a given threshold. Higher moments such as the variance, however, provide complementary information about

the precision and timing errors in cellular events. Our analysis of average behaviour is accompanied by a supplementary examination of variability in first-passage times (see Supplementary Note 4).

In the following sections, we will refer to the time that the deterministic process first reaches a given threshold as the *deterministic FPT*, and we will denote this by $\tau_d$. The expected time for the stochastic process to reach a given threshold given the system is started from $n$ initial molecules will be written as $\langle \tau_n \rangle$. We will write the ratio of the MFPT to the deterministic FPT as $\eta = \langle \tau_n \rangle / \tau_d$.

**The bursty birth–death process.** We consider the bursty birth–death process with fixed burst size $r$,

$$\varnothing \xrightarrow{K} rP, \qquad P \xrightarrow{\delta} \varnothing. \tag{8}$$

See Fig. 2A for an illustration of this model. It is assumed there is the continuous production of a molecular species $P$ in bursts of fixed size $r$, according to a Poisson process at constant rate $K$. The degradation process captures both active degradation as well as dilution due to cell division, and is modelled as a first-order Poisson process with rate $\delta$. Throughout the remainder of the paper, we rescale time as $\tau = \delta \cdot t$, and assume that all model parameters have been scaled by the degradation rate $\delta$, so that $\delta = 1$. Hence both time and the rate parameters are non-dimensional. The bursty birth–death process serves as a crude model for bursty expression of either mRNA or protein due to transcription or translation; we do not here account for the fact that the burst size in these cases is not fixed, but is distributed according to a geometric distribution[11]. For specificity, in what follows we shall refer to $P$ as protein.

**Case 1: the simple birth–death process.** When $r = 1$, the model given in reaction scheme (8) coincides with a simple birth–death process. We are interested in the mean waiting time for the birth–death process to reach a fixed protein number $N$, given that the system is started from $n$ initial proteins. Simulations of the simple birth–death process suggest the MFPT is smaller than the deterministic FPT (see Fig. 2B, but note that only a small sample of trajectories are shown here for visualisation purposes). We now prove this analytically. To begin, we find an analytical solution for the MFPT of the stochastic description of the system, as given by the CME (see Equation (3) of the Supplementary Information).

Starting from Eq. (7), it can be shown (Supplementary Note 2), that the expected waiting time to reach $N$ proteins given the system is started from $n \leq N$, is given by,

$$\langle \tau_n \rangle = \mathbb{E}[\tau_n] = \frac{1}{K}\left( \sum_{i=0}^{N-1} \frac{N^{\underline{i+1}}}{i+1}\left(\frac{1}{K}\right)^i - \sum_{i=0}^{n-1} \frac{n^{\underline{i+1}}}{i+1}\left(\frac{1}{K}\right)^i \right), \tag{9}$$

where for real number $x$ and positive integer $m$, the notation $x^{\underline{m}}$ abbreviates $x(x-1)...(x-(m-1))$, the falling factorial of $x$. In what follows for simplicity we shall set $n = 0$. The deterministic description of the system is given by the reaction rate equation $d_\tau X = K - X$, with $X(0) = 0$, and solving for $X(\tau)$ yields $X(\tau) = K(1 - e^{-\tau})$. Solving for $\tau$ then gives,

$$\tau_d = -\ln\left(1 - \frac{X}{K}\right). \tag{10}$$

Consider some proportion $0 < \rho < 1$, and consider the expected waiting time to reach $\rho K$, that is, some proportion $\rho$ of the steady-state mean. Then Eq. (10) simplifies to

$$\tau_d = -\ln(1 - \rho), \tag{11}$$

which we note is independent of $K$, and where the requirement of $\rho < 1$ enables this to be well defined. Now let us consider Eq. (9) for $N = \lfloor \rho K \rfloor$. For notational simplicity only, we assume that $\rho K$ is an integer, and again observe that the numerator $N^{\underline{i+1}}$ is bounded above by $(\rho K)^{i+1}$. Replacing this in Eq. (9) gives

$$\sum_{i=0}^{N-1} \frac{\rho^{i+1}}{i+1}. \tag{12}$$

Recognising the Taylor expansion of $-\ln(1 - \rho)$ around 0 (for $\rho < 1$), we find that Eq. (12) is the truncated Taylor expansion of Eq. (11). Thus, as each term of the series is strictly positive, it follows that for any proportion $\rho$ of the steady-state mean, the deterministic waiting time is a strict upper bound for the mean waiting times. An example is given in Fig. 2C for $\rho = 0.8$. Here we show how the ratio of the stochastic mean waiting time to the deterministic waiting time, $\eta$, of the simple birth–death process scales with increasing $K$. Note that since $\delta = r = 1$, $K$ is equal to the steady-state mean number of molecules.

**Case 2: burst size $r > 1$.** We now consider the case where protein is translated in bursts of size $r > 1$. Again, we are interested in the waiting time for the bursty system to reach some fixed target $N$, given that the system starts with $n = 0$ initial proteins. In Fig. 2D (top), we consider the case of a high threshold value; here the target $N$ is equal to a large proportion (90%) of the steady-state mean, $Kr$. Noting that the deterministic model will not reach the steady-state mean in finite time (unlike the stochastic model), we expect a faster MFPT as predicted by the stochastic model than that predicted by the deterministic model. Indeed, we see that the stochastic MFPT to reach $N$ (red vertical line) is again smaller than the deterministic FPT (blue vertical line). In Fig. 2E (left), we show how $\eta$ scales with increasing $K$, for a high threshold. The differences are most pronounced for systems with a smaller steady-state mean number of molecules, with waiting times eventually converging in the limit of large mean molecule numbers.

To build intuition for the opposite case, that is, targets that are a small proportion of the steady-state mean, we again look to the extremes. If the target $N$ is chosen such that $N < r$, that is, strictly below the burst size, then the expected waiting time to surpass this in the stochastic case is simply the expected waiting time to the first event, which is $1/K$; this is independent of $N$. But as $N$ tends to 0, the deterministic solution also tends to 0, so that a small enough choice of $N$ will place it faster than $1/K$; the precise boundary for this can be derived as $N < Kr(1 - e^{-1/K}) \approx r$. Note that this condition cannot hold for the simple birth–death process as $r = 1$ and $N$ is discrete. In Fig. 2D (bottom), we see that this difference extends to targets that are much higher than the burst size; here $N = 40$ and $r = 10$. The waiting time as predicted by the deterministic solution (blue vertical line) is indeed smaller than the MFPT to reach $N$ (red vertical line). Comparing the trajectories in Fig. 2D top and bottom, we see that the deterministic solution is faster than the stochastic solution when the target threshold is low, in agreement with our theoretical argument above. In Fig. 2F we show how $\eta$ scales with increasing $K$, for a low threshold value. The differences are most pronounced when the steady-state mean molecule numbers are small.

Here we considered a fixed burst size $r$, however, experimentally a geometric burst size distribution has been observed[60]. We show that these results extend to the case where the burst size is geometrically distributed with mean $b$ (Supplementary Information, Section 5).

**The telegraph model.** Next we investigate whether or not the same transition in the ratio of the deterministic to stochastic waiting times from below to above 1 is possible using a more realistic model of gene expression. We consider the two-state telegraph model of gene expression[61], which has been widely employed in the literature to model bursty gene expression in eukaryotic cells[62,63]; a schematic is

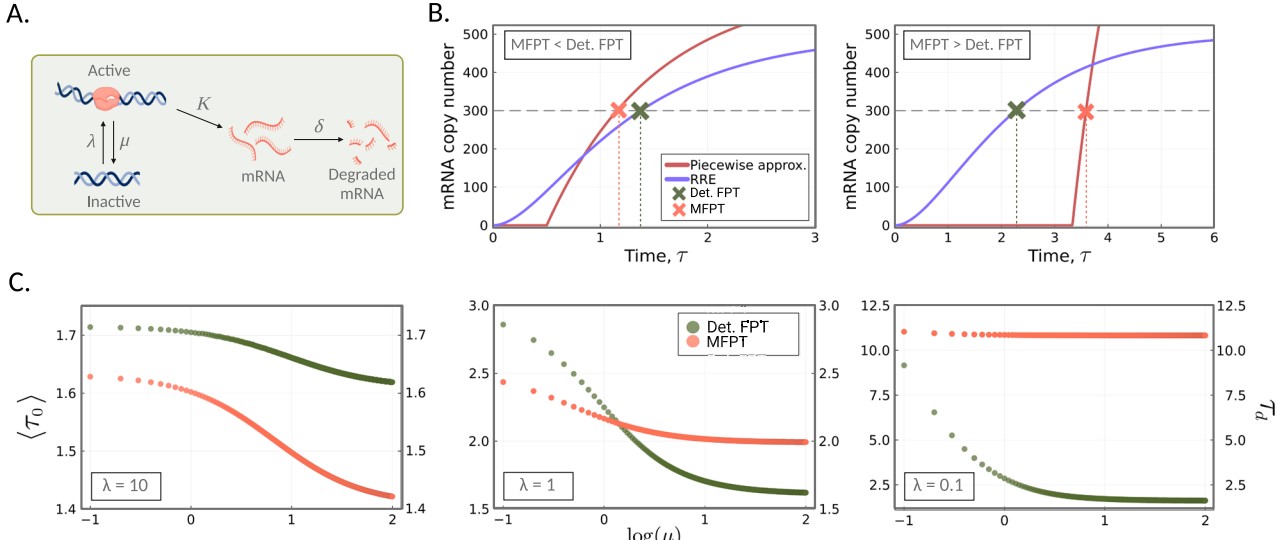

**Fig. 3 | Stochastic *vs.* deterministic waiting times to reach a target mRNA value for the telegraph model. A** A schematic of the telegraph model as given by the reaction scheme in Eq. (13). The schematic was created with BioRender.com. **B** Piecewise-deterministic approximation of the stochastic model (Eq. (14)) versus the deterministic model. (left) For large $\lambda$ values, the approximate MFPT, $\langle \tau_0 \rangle$, (orange cross) of the telegraph process to reach target $N$ (grey dashed line) is smaller than the deterministic FPT, $\langle \tau_d \rangle$, (green cross). The deterministic reaction rate equation is shown in purple. Model parameters are $\lambda = 2$, $\mu = 0.5$, $K = 625$, so that the steady-state mean mRNA copy number is 500. We also have that $\rho = 0.6$, so

that the target $N = 300$. (right) For small values of $\lambda$, the approximate MPFT of the stochastic model (orange cross) is larger than that of the deterministic model (green cross). Model parameters used are $\lambda = 0.3$, $\mu = 0.5$, $K = 1333$, and $\rho = 0.6$. **C** Deterministic FPTs, $\tau_d$ (green), and stochastic MFPTs, $\langle \tau_0 \rangle$ (orange; computed using FSP), as a function of the switching off rate $\mu$, for three different $\lambda$ values: $\lambda = 10$ (left), $\lambda = 1$ (middle), $\lambda = 0.1$ (right). In each plot, we vary $\mu$ from 0 to 100, while keeping the steady-state mean mRNA number fixed at 100; this involves varying $K$ accordingly. The target $N$ is set to 80% of the steady-state mean.

given in Fig. 3A. The effective reactions for the telegraph model are given by,

$$G^* \underset{\mu}{\overset{\lambda}{\rightleftharpoons}} G, \quad G \overset{K}{\longrightarrow} G + M, \quad M \overset{1}{\longrightarrow} \varnothing. \tag{13}$$

In this model, the gene switches probabilistically between an inactive state $G^*$ and active state $G$ from which mRNA molecules ($M$) are synthesised; degradation of mRNA occurs independently of the promoter state.

Again, we are interested in examining waiting times of the telegraph system to reach some target $N = \rho\left(\frac{K\lambda}{\lambda + \mu}\right)$, that is, some proportion $0 < \rho < 1$ of the steady-state mean number of molecules. In contrast to the model considered in the previous section, the gene can now experience periods of inactivity in which no transcription occurs. If the telegraph system is initialised in the inactive state $G^*$, then the waiting time to reach $N$ starting from 0 molecules is *at least* the expected waiting time for the system to transition to the active state (the first event), which is equal to $1/\lambda$. Thus, $1/\lambda$ is a strict lower bound for the mean first-passage time of the system to reach the target $N$.

If we assume that mRNA is abundant enough so that conditional on the gene state the dynamics are deterministic, then the mRNA dynamics, from $\tau = 0$ to $\tau = \tau_{\text{off}}$, are approximated by,

$$f(\tau) = \begin{cases} 0 & \text{if } 0 \leq \tau \leq 1/\lambda, \\ K(1 - \exp(-(\tau - \lambda^{-1}))) & \text{if } 1/\lambda < \tau \leq \tau_{\text{off}}, \end{cases} \tag{14}$$

where $\tau_{\text{off}} = (1/\lambda) + (1/\mu)$ (mean total time for the gene to turn on and then off). Note that the first line describes no expression while the gene is off while the second describes the switching on of gene expression. In particular, for times close to the switching on time, we have that $f(\tau) \approx K(\tau - \lambda^{-1})$, implying there is a sudden linear increase in expression with a slope that is dependent on the mRNA synthesis rate $K$. In contrast, the deterministic model solution is non-zero for all times $t > 0$ because in this case both the gene and mRNA are both treated in a continuous manner. Thus, due to the immediate initiation of mRNA

production in the deterministic model, it consistently maintains a competitive edge over the stochastic model in terms of reaching the target first. However, the stochastic model possesses a notable advantage: upon gene activation, the initiation of mRNA production progresses considerably faster compared to the deterministic model. Because of these two opposing properties, we expect the MFPT to be less than the deterministic FPT if the activation rate is sufficiently large (Fig. 3B left) and the opposite if the activation rate is quite small (Fig. 3B right).

These two scenarios are corroborated by computing the stochastic MFPTs via our FSP approach (Fig. 3C left and right, respectively). Note that in these plots we are varying the switching off rate $\mu$ and synthesis rate $K$ such that the steady-state mean mRNA is unchanged. While the variation of $K$ changes the sharpness of the mRNA response when a gene switches on, its effect on the difference between the MFPT and its deterministic equivalent is secondary to the influence of the switching on rate $\lambda$, provided this is very large or very small. However, when $\lambda$ is moderately large then the value of $K$ becomes the determining factor (Fig. 3C middle). We remark that while the piecewise function Eq. (14) well approximates the mRNA for large and small $\lambda$ values, for intermediate values the theory is unable to account for the fact that a proportion of trajectories will not reach the target on the time interval $0 \leq \tau \leq \tau_{\text{off}}$—hence this approximation cannot be used to accurately predict the value of $\mu$ at which there is a transition in the waiting time ratio from above to below 1 in Fig. 3C middle, and we therefore omit the approximation from the figure.

Furthermore, note that transition in the waiting time ratio $\eta$ seen in the telegraph model is broadly speaking similar in character to that in the simpler bursty birth–death model. Large $\lambda$ means the gene is mostly on and hence this is similar to constitutive expression (the birth–death model with $r = 1$) for which we proved the deterministic FPT to be an upperbound for the MFPT of the stochastic model (Fig. 2C). Small $\lambda$ means the gene is mostly off and if the synthesis rate is large enough this implies bursty ($r > 1$) expression for which the MFPT can be larger than its deterministic counterpart (Fig. 2F). Given

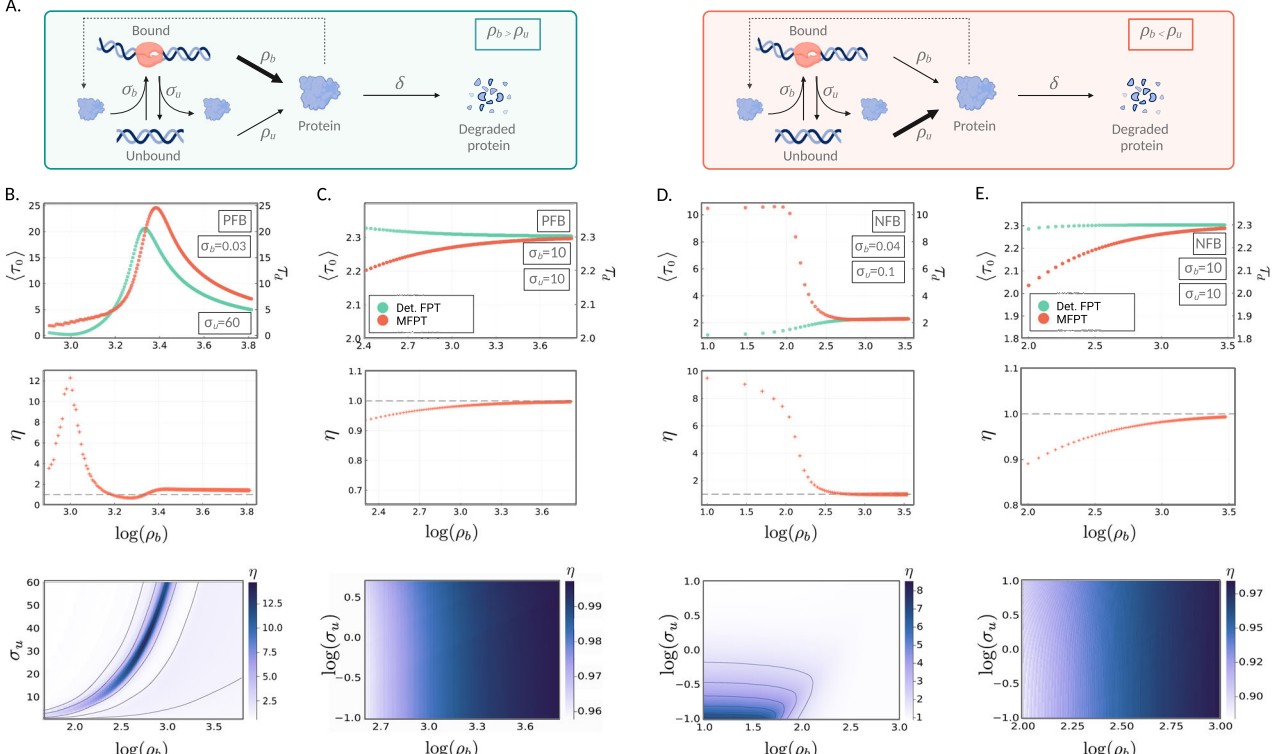

**Fig. 4 | Stochastic *vs.* deterministic waiting times to achieve a target protein value for autoregulatory feedback models. A** (left) An illustration of the feedback loop (given in reaction scheme (15)) for positive feedback (PFB) (i.e., $\rho_u < \rho_b$), and (right) negative feedback (NFB) (i.e., $\rho_u > \rho_b$). The schematic was created with BioRender.com. **B**–**E** Top panel: MFPT, $\langle\tau_0\rangle$, of the stochastic system (orange) to reach 90% of the steady-state mean, given the system is started from 0 protein molecules, for increasing $\rho_b$. The corresponding deterministic waiting times, $\tau_d$, are shown in green. Middle panel: the corresponding ratio, $\eta = \langle\tau_0\rangle/\tau_d$, of the stochastic waiting time to the deterministic waiting time, as a function of $\rho_b$. Bottom panel: ratio, $\eta = \langle\tau_0\rangle/\tau_d$, of the MFPT of the stochastic system to the deterministic FPT, to reach 90% of the steady-state mean, given the system is started from 0 protein molecules, as a function of $\rho_b$ and $\sigma_u$. All parameters along with the mean molecule numbers are given in Supplementary Table 1.

the qualitative similarities in the predictions of the two models, it is likely the form of the burst size distribution (a delta function for the bursty birth–death model and a geometric distribution for the telegraph model in certain limits[12]) is not important to the existence of the observed transition in $\eta$.

**An autoregulatory feedback loop.** Next we examine an autoregulatory feedback loop in which a protein produced by a gene either enhances or suppresses its own expression. This motif is commonly encountered in biology. For example, in *E. coli* negative autoregulation appears in over 40% of known transcription factors. A number of studies have used stochastic models to investigate optimal feedback mechanisms to minimise the variability in FPTs[48,50,64–68]. The reaction scheme for the feedback loop that we employ is given by Grima et al.[69],

$$P + G \underset{\sigma_u}{\overset{\sigma_b}{\rightleftharpoons}} G^*, \quad G \xrightarrow{\rho_u} G + P,$$
$$P \xrightarrow{1} \varnothing, \quad G^* \xrightarrow{\rho_b} G^* + P, \tag{15}$$

where $G^*$ and $G$ represent the bound and unbound states, respectively, and $P$ represents protein. The parameters $\sigma_u$ and $\sigma_b$ are the unbinding and binding rates, respectively, and $\delta$ is the protein degradation rate. The rate of protein production depends on the gene state and is given by $\rho_b$ and $\rho_u$ in the bound and unbound states, respectively. When $\rho_b > \rho_u$ the feedback is positive since an increase in the protein number will cause the gene to switch more often to the $G^*$ state in which the production rate is high. Similarly when $\rho_b < \rho_u$ the feedback is negative. We illustrate the positive and negative feedback loops in Fig. 4A left and right panels, respectively. More realistic models have been devised, for example[70] describes a bursty version of reaction scheme

(15) to account for translational bursting; however, these details are unlikely to change the qualitative properties of the waiting time results that we will describe next.

In Fig. 4B, C (vertical panels), we consider the case of positive feedback (i.e., when $\rho_u < \rho_b$). For two different $\sigma_b$ values, we show the MFPT of the stochastic system (orange) to reach 90% of the steady-state mean, given the system is started from 0 protein molecules, for increasing $\rho_b$; this can be thought of as increasing the strength of the positive feedback. The corresponding deterministic waiting times are displayed in green. For low values of the binding rate $\sigma_b$ (Fig. 4B), we see a rapid rise and fall in the FPTs (for both the stochastic and deterministic models), as we increase the strength of positive feedback. This initial rise in the waiting time may seem somewhat counterintuitive, but can be understood in terms of the corresponding increase in the steady-state mean. As the transcription rate $\rho_b$ rises, so too does the steady-state mean, and hence the target $N$. Because $\sigma_b$ is very low, binding is a rare event and short lived, as $\sigma_u$ is large. This means that in a typical trajectory, most of the time towards reaching the target is spent in the unbound state with constant transcription rate $\rho_u$, attempting to reach a rising target. For high values of $\rho_b$, the level of transcription is sufficiently high that even the short window of relatively rare bound behaviour provides significant progress towards achieving the target molecule number. For high $\sigma_b$ values (Fig. 4C), the effective binding rate, that is $\sigma_b\langle P\rangle$, quickly becomes enormously higher than the unbinding rate $\sigma_u$, so that the system is very close to constitutive, and the deterministic FPTs bound from above the stochastic MFPTs. The corresponding waiting-time ratio plots also reveal that this ratio can be quite large when it is above 1 (an order of magnitude)—this is a considerably larger effect than seen for the bursty

birth–death and telegraph models, likely because feedback models have a bimolecular reaction which enhances the differences between the SSA trajectories and the deterministic models[59]. Note that the largest differences in the deterministic and stochastic waiting times occur within the range of tens to hundreds of molecules (here the mean number of molecules is approximately between 11.68 and 370.91). This is within the range known for many cells. For example, in *E. coli*, copy numbers of transcription factors are quite low, occurring between 1 and 1000 per cell[71,72]. The heatmaps in Fig. 4B, C (bottom plots) show that the behaviour in the top and middle plots are qualitatively representative across a broad range of $\sigma_u$ values.

In the case of negative feedback (i.e., when $\rho_b < \rho_u$), increasing $\rho_b$ now leads to a decrease in the strength of negative feedback, and hence again a rise in the overall mean steady-state protein number. Thus, the overall trends are broadly similar to the positive feedback case. For a low $\sigma_b$ value (see Fig. 4D with $\sigma_b = 0.04$), we initially see a very slight increase in the MFPT in the region of low $\rho_b$. This is followed by a rapid decline in the MFPT, with high values of $\rho_b$ (that is, approaching equality with $\rho_u$), becoming close to a constitutive model. We note that again the largest differences in the stochastic and deterministic waiting times occur within the range of tens to hundreds of molecules (between approximately 44.36 and 317.1). For high $\sigma_b$ (Fig. 4E with $\sigma_b = 10$), the effective binding rate is significantly higher than $\sigma_u$, so that the system is close to being constitutive with protein production rate at $\rho_b$. Hence the stochastic MFPTs are bounded above by the deterministic FPTs; refer to Fig. 4E (top and middle). The heatmaps in Fig. 4D, E (bottom) again show that the parameter sets in D, E (top and middle) are qualitatively representative across a broad range of $\sigma_u$ values.

## Discussion

In this article, we have investigated the differences between deterministic and stochastic model predictions for the timing of cellular events, when the target molecular value can be reached by the deterministic model in finite time. While differences between the mean molecule number predictions of these models have previously been extensively investigated[59,73], differences in their predictions of the mean time to trigger a cellular event have not.

Unlike previous studies[13,46–52] that have focused on the parametric dependence of MFPTs, here the focus has been on the interplay between noise and dynamics on the mean trigger time. The qualitative differences in MFPTs of deterministic and stochastic models, as observed here, were indeed surprising. Although the importance of stochasticity in gene expression is nowadays well recognised, there is often still an expectation or assumption that analysis of mean quantities can be done using deterministic frameworks. Our findings show that the MFPT is one of the features of stochastic dynamical systems for which this assumption may not be valid.

For the systems that we have investigated in detail here—birth–death processes, the telegraph model, and an autoregulatory feedback loop—we find quantitative as well as qualitative differences between deterministic and stochastic dynamics. For the simple birth–death process, the deterministic model prediction provides a strict upper bound for the MFPT predicted by the stochastic model. In other words, intrinsic noise leads to a reduction of the trigger time—this effect becomes more appreciable as the mean steady-state molecule number decreases. For the bursty birth–death process, there is a switch from a regime where intrinsic noise reduces the mean trigger time to one where it lengthens it, as the target molecular value is decreased. Since the expression of many genes is bursty (of transcriptional or translational origin), this simple model suggests that the transition may exist in more realistic models of gene expression. Indeed, we showed that the same transition occurs in the telegraph model, which is capable of explaining both constitutive, bursty, and

intermediate behaviour, and in a model of autoregulatory feedback, a common regulatory motif in nature.

For feedback loops, we observed significantly higher ratios of deterministic to stochastic FPTs compared to previous models. This has implications in the timing of developmental processes governed by molecular clocks, where dynamics are often modelled deterministically, e.g. circadian clocks in *Drosophila*[74,75] and *Neurospora*[76,77], and segmentation clocks in vertebrates[78–82]. Additionally, our results provide insights into the non-monotonic dependence observed in both the deterministic FPT and MFPT on the feedback strength. Similar non-monotonic behaviour of the MFPT has been shown in stochastic threshold models[83].

The effect of feedback on the differences between the predictions of the deterministic and stochastic frameworks for the mean molecule numbers are well known to increase with decreasing system size (volume of the system), and with increasing proximity to the point in parameter space where a system switches from stable to unstable dynamical behaviour[59]. In fact, our analysis of the waiting times for the substrate to reach a certain target level in an enzyme-substrate reaction (Fig. S11) showed that the ratio of deterministic and stochastic mean waiting times also increased with these two system properties.

Concluding, our study shows that intrinsic noise has non-trivial effects on the timing of cellular events and that simple models can provide intuitive insights into the microscopic origins of these effects. The hope, whether implicit or explicit, that deterministic models predict accurate mean quantities appears misplaced for MFPTs which has important implications for modelling-based studies of biochemical timing.

## Methods
### Stochastic reaction networks
Consider a well-stirred mixture consisting of $N$ chemical species $S_1, \ldots, S_N$ that interact through $M$ chemical reactions $R_1, \ldots, R_M$,

$$R_j \equiv \sum_{i=1}^{N} s_{ij} x_i \xrightarrow{c_j} \sum_{i=1}^{N} r_{ij} x_i, \quad j \in \{1, \ldots, M\}, \tag{16}$$

where $x_i$ denotes the number of $S_i$ molecules in the system at time $t$, $s_{ij}$ and $r_{ij}$ are integers, and $c_j$ is the rate constant of reaction $R_j$ with units of inverse time. Throughout, we will let $\mathbf{x}(t) = (x_1(t), \ldots, x_N(t))$ represent the state of the system at time $t$. Each reaction $R_j$ has an associated *propensity function* $a_j$ given by,

$$a_j(\mathbf{x}) = c_j h_j(\mathbf{x}), \tag{17}$$

where $h_j(\mathbf{x})$ is defined to be the number of distinct combinations of $R_j$-reactant molecules available in the state $\mathbf{x}$. The *state-change*, or *stoichiometry vector* $\mathbf{v}_j$ is defined to be the vector $(v_{1j}, \ldots, v_{Nj})$ whose $i$th component is given by $v_{ij} = r_{ij} - s_{ij}$, for $i \in \{1, \ldots, N\}$ and $j \in \{1, \ldots, M\}$. The process $\mathbf{x}(t)$ is a continuous-time Markov process, and the time evolution of the joint probability distribution of the molecule numbers is described by the Chemical Master Equation (CME),

$$d_t \mathbf{P} = \mathbf{A}\mathbf{P}, \tag{18}$$

where $\mathbf{P} := [P(\mathbf{x}_1), P(\mathbf{x}_2), \ldots]^T$ and $\mathbf{A}$ is the state transition matrix with the following structure[84],

$$A_{ik} := \begin{cases} -\sum_{j=1}^{M} a_j(\mathbf{x}), & \text{for } i = k \\ a_j(\mathbf{x}_i), & \text{for all } k \text{ such that } x_k = x_i + \mathbf{v}_j \\ 0, & \text{Otherwise} . \end{cases} \tag{19}$$

### Finite State Projection for the modified CME
Numerical computation of first-passage time (FPT) distributions can be performed using an adaptation of the Finite State Projection (FSP)[41].

Here the state space of the stochastic reaction network, which is usually infinite, is reduced to a finite subset consisting of the most relevant states. This converts the CME into a finite linear system of equations that can be solved efficiently on a computer.

In the standard formulation of the FSP, we select a finite set $\mathbf{Z}$ of states, which we assume to include the initial set, as well as the target set $\mathbf{Y}$. The terms in the CME corresponding to any state outside of $\mathbf{Z}$ are further assumed to vanish. Mathematically, this corresponds to solving the exact CME for a modified reaction network, wherein we combine all states outside of $\mathbf{Z}$ into a single state $\mathbf{Z}^c$ and remove all transitions from $\mathbf{Z}^c$ to $\mathbf{Z}$. Comparing this with the matrix $\mathbf{A}_{\mathbf{Y}}^T$, which contains all states excluding those in $\mathbf{Y}$, we see that applying the FSP to the construction of the matrix $\mathbf{A}_{\mathbf{Y}}$ eliminates all states that are either outside of the truncation, or in the absorbing set $\mathbf{Y}$. That is, we compute exactly the FPT distribution until the system either enters $\mathbf{Y}$ or leaves the truncated state space.

As a result, numerically estimating FPTs using the FSP will always underestimate the true first passage times, since the system leaving the truncated space is treated the same as the system entering the absorbing set $\mathbf{Y}$. To minimise the approximation error, therefore, one should choose the truncation $\mathbf{Z}$ such that the system is unlikely to leave $\mathbf{Z}$ before entering the target set $\mathbf{Y}$.

To compute FPTs numerically, we use the `FiniteStateProjection.jl` package[85] to construct the matrix $\mathbf{A}_{\mathbf{Y}}$, and solve the system of equations (Eq. (7)) using the standard sparse solvers provided in Julia.

### Reporting summary
Further information on research design is available in the Nature Portfolio Reporting Summary linked to this article.

## Data availability
This research paper does not contain any empirical data. The results presented here are based on simulated data generated through computational models, implemented in Julia.

## Code availability
Detailed descriptions of the code used for the analysis in this paper are available in a GitHub repository which can found here https://zenodo.org/doi/10.5281/zenodo.11201704, along with instructions for replicating the analysis.

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

## Acknowledgements

We gratefully acknowledge support from the members of the *Theoretical Systems Biology Group* at the University of Melbourne. L.H. and M.P.H.S. are supported by the University of Melbourne DVCR fund, M.A.C. by the University of Melbourne graduate research scholarship, K.Ö. by the EPSRC Centre for Doctoral Training in Data Science (EPSRC grant EP/L016427/1) and R.G. by the Leverhulme Trust grant (Grant No. RPG-2020-327). L.H. and M.P.H.S. acknowledge funding through an Australian Research Council Discovery Project (DP220101005) and the Australian Research Council Centre of Excellence for the Mathematical Analysis of Cellular Systems (CE230100001); K.Ö. and M.P.H.S. acknowledge funding through an Australian Laureate Fellowship (FL220100005).

## Author contributions

L.H., M.A.C., K.Ö., R.G. and M.P.H.S. conceptualised the research. Formal analysis and visualisation was conducted by L.H., M.A.C. and K.Ö. Writing by L.H. and M.A.C., with editing and review contributed by K.Ö., R.G. and M.P.H.S. All authors provided critical feedback and helped shape the research.

## Competing interests

The authors declare no competing interests.
