## [Peer Review File · Nature Communications]

A stochastic vs deterministic perspective on the timing of cellular eventsREVIEWER COMMENTS

Reviewer #1 (Remarks to the Author):

The authors provide a theoretical investigation of the first passage times (FPT) of stochastic gene regulatory circuits. Specifically, they provide a theoretical method to compute the mean FPT. The results are significant for understanding the precision of timing in cellular events and how it can be regulated. The authors find that the mean FPT can be shorter or longer when low copy number conditions. The manuscript is well-written and thought-provoking.

The main limitation of this work is that only the mean is considered while the variance is ignored. The latter would be informative of timing errors and precision of cellular events. For example, it could be that under realistic conditions the FPT is very wide such that the difference between deterministic and stochastic mean FPTs is negligible compared to the noise. Hence it is an important point that the authors should address. I have several other comments that I like to see addressed:

-- Is there a better way to quantify how bad the deterministic FPT approximation is compared to MFPT? Seems like the variance in the FPT distribution should somehow factor into this. Are there examples where the deterministic model predicts an FPT well outside of where the bulk of the FPT distribution of the stochastic model lies? If the FPT distribution is very noisy then neither deterministic FPT and MFPT seem very informative. See my comment above.

-- A model system where event timing has been quantitatively linked to gene expression noise is lysis time variation in the bacteriophage lambda as mentioned in the intro. Can the theory provide further insights to this problem? See for example, <https://royalsocietypublishing.org/doi/full/10.1098/rsif.2014.0140>

-- Is the method to compute the FPTs new or has this been considered before? How does the method extend to FPT statistics over a distribution of initial states?

-- Section 3.1. The bursty birth-death process is restricted to constant bursts. Realistically, geometrically distributed bursts should be considered.

-- Figure 3. Panel B shows the piecewise approximation of the FPT. This seems to be omitted from comparisons in Panel C?

-- Figure 3. The way the model is parametrised with the protein production rate several orders of magnitude higher than the gene switching rates seems to allude to the variance in the FPT distribution being dominated by the switching time. What does the resulting first passage time distribution look like? Is the mean a useful characterisation of that distribution to begin with?

-- Discussion. It is pointed out that the agreement between the deterministic FPT and MFPT occurs in the limit of large molecule numbers while for the telegraph model, it is not the case. I think a bit of care should be taken here. Both the telegraph model and the autoregulatory feedback loop feature a counting variable for on and off state of the gene that are going to take values 0 and 1. It is clear that the deterministic approximation relying on large copy numbers will be no good here. A piecewise deterministic Markov process approximation where the gene switch is kept stochastic would be a much better approximation.

Reviewer #2 (Remarks to the Author):

This paper compares event timing in stochastic versus deterministic models of biochemical reactions. First passage times (that is, the time taken to first cross some threshold amount for a molecular species in the model) are compared for a set of common small models of gene regulation and catalysis (and, in the supplement, an epidemic SIR-type (non-biochemical) model).

The methodology of the paper is clear and sound and the results are likely valuable and interesting to the field. Several of the results/models involve stochastic gene expression, and these types of simple models have steadily gained traction in recent years with interest in gene expression noise and its biological consequences.

The relationship between the stochastic and deterministic predictions of First Passage Times (FPTs) was interesting, as it depended significantly on the type of model and parameter regime. Some of the results were not intuitive (particularly Figure 4B, which shows non-monotonic behavior of the waiting time ratio for the autoregulatory circuit). And, as the authors point out, obtaining these comparisons directly by simulation would have been challenging, without the aid of their Master Equation-based framework.

It could strengthen (in my view) the significance of the paper if some general principles could be uncovered by the study, but perhaps no simple, generalizable principles exist. The only general "takeaway" that I saw (in addition to the finding that deterministic models are often bad at predicting the "true" stochastic mean FPT) was that feedback (e.g., in the autoregulatory circuit) increases the discrepancy. But this was only shown for one type of feedback model, though both positive and negative feedback were studied.

The authors make the case that the discrepancy between stochastic and deterministic results are important because deterministic models are usually the standard in biochemical modelling. Therefore, reliance on deterministic models alone could lead to poor predictions. While this is true, I am not sure of examples where biochemical deterministic models have been relied on alone for event timing (though likely such examples exist). In contrast, many studies have looked at applications of stochastic FPT analysis (as cited by the authors, on page 2). It could strengthen the significance of this study if the authors could point to specific cases in which deterministic models were used for event timing, but in which the stochastic prediction would be different. Specific comments/suggestions.

1. In the Summary, this sentence:

"For the birth death process, this agreement occurs only in the limit of large molecule numbers, while for the telegraph model not even then do they agree. "

I did not find results in the paper that supported the latter part of this sentence. The statement that, in the limit of large molecule numbers, the stochastic MFPT does not agree with the deterministic result is surprising, since one would expect that any observable, including MFPT, should agree in the large-number limit. In Figure 3 (showing results for the telegraph model), to my understanding the mean mRNA number is not changing, so there is no figure that compares the stochastic versus deterministic passage time as a function of system size. The sentence should be modified or explained.

2. A small thing, but I found it slightly curious that enzyme-mediated catalysis was mentioned in the abstract, but these results were not shown or discussed in the main text anywhere. It would be helpful to at least summarize the results for Michaelis-Menten model somewhere in the main text, and explain how those results compare with those of the main-text models.

Reviewer #3 (Remarks to the Author):

Reviewer #4 (Remarks to the Author):

The manuscript analyses first passage time (FTP) problems relevant to various biochemical processes and exemplified by models for gene expression. In particular, the authors show that for three prototypical systems (bursty birth-death process, telegraph model, autoregulatory feedback

loop) the mean FPT (MFPT) of the stochastic system can either be smaller or larger than the deterministic time to reach a given threshold. Overall, the manuscript is well written and provides convincing support for the authors' claims.

- I completely agree with the authors about the importance of stochastic models for understanding cellular events, and that deterministic differential equations have been, and still are, used predominantly for mathematical ease despite the overwhelming and still accumulating evidence for small numbers of molecules involved in biochemical reactions. There is, however, one conceptual issue that I would like the authors to make much more explicit in their introduction. By comparing stochastic and deterministic dynamics, there is an implicit assumption that writing down the deterministic equations is conceptually valid even if results differ between the two approaches. However, this is not true. The Law of Mass action, which is used for deriving the deterministic equations, does not hold when molecule numbers are small. In that sense, the comparisons in the manuscript amount to comparing an appropriate model to an invalid one. This point does not take away from the excellent work done in the manuscript, but there should be a clear statement regarding the appropriateness of models.

- That cellular decisions can be modelled as FPT problems is undoubtedly true. In light of this, the introduction could benefit from a wider selection of examples that go beyond the standard gene expression and cell cycle illustrations. For instance, activation of complex molecules, which then trigger downward signals, can be conceptualised as a FPT problem. Another example is the formation of a critical nucleus that triggers an intracellular calcium wave. The latter also shows that FPT problems occur in physiologically relevant spatially extended systems. Relevant papers are

- R. Thul, M. Falcke, Waiting time distributions for clusters of complex molecules, *Europhysics Letters*, 79, 38003 (2007)
- R. Thul, K. Thurley, M. Falcke, Toward a predictive model of Ca²⁺ puffs, *Chaos*, 19, 037108 (2009)
- L. Ramlow, M. Falcke, B. Lindner, An integrate-and-fire approach to Ca²⁺ signaling—The noise of puffs, *Biophysical Journal*, 122 (3), 237a (2023)
- S. Rüdiger, Stochastic models of intracellular calcium signals, *Physics Reports* 534 (2), 39-87 (2014)

- A main emphasis of the manuscript is the role of noise on the MFPT. A non-monotonic dependence of the MFPT on the noise strength was shown in W. Braun, P. C. Matthews, R. Thul, First-passage times in integrate-and-fire neurons with stochastic thresholds, *Physical Review E*, 91, 21953 (2015), where the MFPT exceeds the deterministic value for small noise strength and then becomes smaller for larger noise strength. The paper also contains analytical results for computing the full FPT distribution.

- In Figure 1, the y-axis of the FPT distribution is slightly misleading. I recommend to move the origin of the coordinate system for the FPT distribution to align with the coordinate system of the time course. In this way, the full FPT distribution can be shown, highlighting that it is zero at $t=0$ and positively skewed.

- On p. 3, the authors write "we are interested in the distribution of first-passage times". Since the paper focusses on MFPT and Section 2.1 only considers moments, I suggest to reword the above sentence as e.g. "we are interested in first-passage times".

- When introducing Y , it might be worth stating that the state-space is discrete (as the entire derivation could be done for a continuous state space as well, but the derivation is only written for a discrete state space). After Equation (1), I would say $z \notin Y$ for completeness. I would like to add that I enjoyed the new proof for the moments of the FPT distribution.

RESPONSE TO REVIEWERS FOR “THE TIMING OF CELLULAR EVENTS: A STOCHASTIC VS DETERMINISTIC PERSPECTIVE”

LUCY HAM, MEGAN A COOMER, KAAAN ÖCAL, RAMON GRIMA, MICHAEL P. H. STUMPF

We were delighted with the reviewers’ enthusiastic and constructive comments and suggestions. We have tried to address the raised points comprehensively, and we feel that this additional effort has improved the clarity and content of the manuscript.

REVIEWER 1 COMMENTS TO THE AUTHOR WITH RESPONSES:

- (1) *The main limitation of this work is that only the mean is considered while the variance is ignored. The latter would be informative of timing errors and precision of cellular events. For example, it could be that under realistic conditions the FPT is very wide such that the difference between deterministic and stochastic mean FPTs is negligible compared to the noise. Hence it is an important point that the authors should address.*

- To address this point, we have included a statement emphasising the significance of considering higher moments (see the first paragraph of Section 3 in the main text), as well as an analysis of the variability in FPTs for all models featured in the main text (refer to Section 5 of the Supplementary Material).

We thank the reviewer for raising this point, and agree that valuable information can be provided by higher moments such as the variance. It is, however, important to recognise that the mean still holds significant utility in statistical analysis, especially in the context of cellular event timing. Specifically, the mean provides insights into the average behavior of these events, which is of primary interest in many (if not all) event timing analyses. Even if the difference between deterministic FPT and stochastic MFPT is negligible compared to the noise, understanding the expected FPT remains essential for predictive modelling and system characterisation.

- (2) *Is there a better way to quantify how bad the deterministic FPT approximation is compared to MFPT? Seems like the variance in the FPT distribution should somehow factor into this. Are there examples where the deterministic model predicts an FPT well outside of where the bulk of the FPT distribution of the stochastic model lies? If the FPT distribution is very noisy then neither deterministic FPT and MFPT seem very informative.*

- Considering examples where the deterministic model predicts a FPT well outside of the bulk of the FPT distribution of the stochastic model is highly valuable to our study, and we appreciate the reviewer’s insight in prompting this investigation. As mentioned in the introduction of the manuscript, if for a given initial condition, the target molecule number is outside of the range predicted by the deterministic model in finite time, then clearly the latter’s prediction for the mean time is undefined, whereas a stochastic model will typically predict a finite value. In these cases, the deterministic model only approaches the target asymptotically. Thus, by choosing a target that is very close to the steady-state mean number of molecules, we expect the FPT prediction of the deterministic model to be well outside of the bulk of the FPT distribution of the stochastic model. We have incorporated an example of this into our analysis for the specific case of a birth-death process. We note, however, that the result is independent of the choice of model (refer to Figure 6 of the Supplementary Material).

We agree that if there is significant variability in the FPT distribution, then it may be the case that neither the deterministic FPT nor the MFPT will offer a sufficiently informative representation of the underlying system. In such cases, metrics beyond FPT statistics may be required to assess the accuracy of the deterministic model. That being stated, significant stochasticity in the system, while contributing to the variance in FPTs, does not necessarily imply systematic inaccuracies or accuracies in the deterministic FPT approximation compared to the MFPT. To truly gauge this, a direct comparison

is needed. Examining the relative difference in the two FPT measures, across different scenarios (as done in our study), offers a clear indication of the accuracy of the deterministic FPT approximation compared to the MFPT.

(3) *A model system where event timing has been quantitatively linked to gene expression noise is lysis time variation in the bacteriophage lambda as mentioned in the intro. Can the theory provide further insights to this problem? See for example, <https://royalsocietypublishing.org/doi/full/10.1098/rsif.2014.0140>.*

► Yes; this system, too, is amenable to simulation and analysis with the tools presented here. Our approach can be used to extend the analysis in the above mentioned paper to compute the full FPT distributions of the stochastic models considered there (see an example here). The relationship between gene expression noise and event timing in the context of lysis times (LTs) can be explained quantitatively by inspection of the FPT distributions, providing further insights into the mechanisms that lead to the variation in LTs. For example, as the gene expression noise in the underlying model is modulated, does the peak of the FPT distribution move towards shorter or longer FPTs times? Are the tails of the FPT distribution heavier, providing insights into frequency of rare long-time events as a function of noise?

(4) *Is the method to compute the FPTs new or has this been considered before? How does the method extend to FPT statistics over a distribution of initial states?*

► To clarify, we emphasise that an existing approach based on the Backward Chemical Master Equation (BCME) can be used to derive our central result (Equation 7 of the main text); this is acknowledged in the main text (see the final paragraph of Page 4), and a full derivation using the BCME is given in the Supplementary Material (see Section 2.1). We note however that the BCME approach is not widely known. In the manuscript, we present a fresh proof of this result that is more direct and intuitive. Additionally, we provide a new computational framework, leveraging a modified FSP algorithm, which enables the computation of FPT moments of any order across a set of initial states. The advantage of our approach is that this can be done simultaneously for *all* initial states, significantly speeding up computational time.

While the primary focus of our study is on the mean FPT, it is worth mentioning that our methodology straightforwardly extends to a time-dependent approach for computing the full FPT distribution over a set of initial states. We have incorporated this extension into the Supplementary Material (see Section 2.2).

(5) *Section 3.1. The bursty birth-death process is restricted to constant bursts. Realistically, geometrically distributed bursts should be considered.*

► In the manuscript, we adopt a fixed burst size for simplicity and ease of proof. We argue that the results are likely not dependent on the form of the burst size distribution (refer to Paragraph 3 of Page 8 in the main text). In fact, by extending our analysis to geometrically distributed burst sizes, we observe qualitatively similar results to the case of fixed burst sizes, and we have now, for completeness sake, included this in the Supplementary Material (see Section 5).

(6) *Figure 3. Panel B shows the piecewise approximation of the FPT. This seems to be omitted from comparisons in Panel C?*

► We acknowledge that the omission of the piecewise deterministic approximation of the telegraph process from Panel C may seem puzzling initially. To clarify, the approximation is meant to serve as a conceptual aid for understanding the early-time kinetics (and therefore FPTs) in specific cases of the telegraph model. As emphasized in the main text (see second paragraph of Page 8), this approach becomes inadequate in intermediate cases. To better elucidate its omission in Panel C, we have added a note in the main text for clarification; refer to Page 8.

- (7) *Figure 3. The way the model is parameterised with the protein production rate several orders of magnitude higher than the gene switching rates seems to allude to the variance in the FPT distribution being dominated by the switching time. What does the resulting first passage time distribution look like? Is the mean a useful characterisation of that distribution to begin with?*
- ▶ We thank the reviewer for drawing our attention to this point, and agree that more clarity was needed regarding the usefulness of the mean in these cases. We have now provided full (time-dependent) FPT distributions in three representative cases for high, intermediate, and low values of λ (refer to the ridgeline plot given in Figure 4(B) of the Supplementary Material). Additionally, we include the corresponding noise, as measured by the coefficient of variation (CV), for each distribution. The results reveal low ($CV \approx 0.27$) to moderate ($CV \approx 0.68$) noise in the distribution for high and intermediate λ . While the noise is higher ($CV \approx 1$) for the case of low λ . The level of noise, therefore, is low enough for the mean to be a reasonably useful characterisation here.
- (8) *Discussion. It is pointed out that the agreement between the deterministic FPT and MFPT occurs in the limit of large molecule numbers while for the telegraph model, it is not the case. I think a bit of care should be taken here. Both the telegraph model and the autoregulatory feedback loop feature a counting variable for on and off state of the gene that are going to take values 0 and 1. It is clear that the deterministic approximation relying on large copy numbers will be no good here. A piecewise deterministic Markov process approximation where the gene switch is kept stochastic would be a much better approximation.*
- ▶ We thank the reviewer for this comment, as it has prompted us to better clarify the statement regarding the telegraph model in the conclusion. Upon reflection, we can see how this statement is ambiguous: as we retain only one gene copy, we agree that we don't technically consider the telegraph model in the large copy number limit. We have now removed this statement accordingly, among other more extensive changes to the discussion. We feel this additional effort has improved the manuscript.

REVIEWER 2 COMMENTS TO THE AUTHOR WITH RESPONSES:

- (1) *It could strengthen the significance of this study if the authors could point to specific cases in which deterministic models were used for event timing, but in which the stochastic prediction would be different.*
- ▶ Circadian clocks provide a compelling illustration of this. These molecular clocks are known to regulate the timing of developmental processes in various organisms, e.g. *Drosophila* [1, 2], *Neurospora* [3, 4], and vertebrates [5], and originate from the autoregulation of so-called clock genes and interlocked feedback loops. These regulatory dynamics are often modelled deterministically by way of delayed differential equations or coupled ordinary differential equations [6, 7, 8, 9]. However, in cellular conditions where the involved molecules are small, stochastic modelling is more appropriate [10]. Our results for autoregulatory models in the current paper highlight notable differences in waiting time predictions between stochastic and deterministic models. We have included the above examples in the conclusion of the revised manuscript to highlight this point. Refer to Page 11 and references 75-83 of the revised manuscript.
- (2) *In the Summary, this sentence: “For the birth death process, this agreement occurs only in the limit of large molecule numbers, while for the telegraph model not even then do they agree. “ I did not find results in the paper that supported the latter part of this sentence. The statement that, in the limit of large molecule numbers, the stochastic MFPT does not agree with the deterministic result is surprising, since one would expect that any observable, including MFPT, should agree in the large-number limit. In Figure 3 (showing results for the telegraph model), to my understanding the mean mRNA number is not changing, so there is no figure that compares the stochastic versus deterministic passage time as a function of system size. The sentence should be modified or explained.*
- ▶ We agree that clarification is required to reconcile the sentence about the telegraph model in the limit of large molecule numbers, and the latter part of the sentence has now been removed. Please refer to

the third paragraph of the discussion in the revised version of the manuscript.

- (3) *A small thing, but I found it slightly curious that enzyme-mediated catalysis was mentioned in the abstract, but these results were not shown or discussed in the main text anywhere. It would be helpful to at least summarize the results for Michaelis-Menten model somewhere in the main text, and explain how those results compare with those of the main-text models.*
- We thank the reviewer for drawing our attention to this. We have incorporated a number of changes into the discussion of the revised manuscript, including a summary of the results for the Michaelis-Menten model, and relate these results to those of the models featured in the main text. Please refer to the discussion for these changes.

REVIEWER 3 AND 4 COMMENTS TO THE AUTHOR WITH RESPONSES:

- (1) *By comparing stochastic and deterministic dynamics, there is an implicit assumption that writing down the deterministic equations is conceptually valid even if results differ between the two approaches. However, this is not true. The Law of Mass action, which is used for deriving the deterministic equations, does not hold when molecule numbers are small. In that sense, the comparisons in the manuscript amount to comparing an appropriate model to an invalid one. This point does not take away from the excellent work done in the manuscript, but there should be a clear statement regarding the appropriateness of models.*
- This is indeed an important point: the appropriateness of the deterministic equations requires critical consideration when molecule numbers are small. It is important to recognise the challenge in defining the precise boundary of validity for the deterministic equations. The difference between predictions of the deterministic and stochastic models, of course, vanish in the limit of large molecule numbers [11]. However, determining exactly when molecule numbers become too small is not always clear.
- Additionally, it is worth noting that deterministic models find wide spread use in the literature, mostly due to their ease of interpretation, and often without rigorous justification. Our paper advocates for a more thorough treatment when dealing with stochastic systems in scenarios characterised by low molecule numbers, and we hope that a clear statement regarding the appropriateness of the deterministic equations will help contribute to a nuanced interpretation of our results. Please refer to the second paragraph of Page 2 in the revised manuscript for these changes.
- (2) *That cellular decisions can be modelled as FPT problems is undoubtedly true. In light of this, the introduction could benefit from a wider selection of examples that go beyond the standard gene expression and cell cycle illustrations. For instance, activation of complex molecules, which then trigger downward signals, can be conceptualised as a FPT problem. Another example is the the formation of a critical nucleus that triggers an intracellular calcium wave. The latter also shows that FPT problems occur in physiologically relevant spatially extended system. Relevant papers are:*
- R. Thul, M. Falcke, *Waiting time distributions for clusters of complex molecules*, *Europhysics Letters*, 79, 38003 (2007)
 - R. Thul, K. Thurley, M. Falcke, *Toward a predictive model of Ca²⁺ puffs*, *Chaos*, 19, 037108 (2009)
 - L. Ramlow, M. Falcke, B. Lindner, *An integrate-and-fire approach to Ca²⁺ signaling—The noise of puffs*, *Biophysical Journal*, 122 (3), 237a (2023)
 - S. Rüdiger, *Stochastic models of intracellular calcium signals*, *Physics Reports* 534 (2), 39-87 (2014)
- The inclusion of a wider selection of examples of FPT problems is an excellent suggestion and we thank the reviewers for highlighting the above works; these will likely be of interest to our intended readership. The spatial FPT example involving a triggering event of an intracellular calcium wave, in particular, has captured our attention. The above listed references have now been incorporated into the introduction, along with some accompanying text to highlight the significance of these examples; please refer to the

second paragraph of the introduction and references 26, 27, 28, and 29 of the revised manuscript.

- (3) *A main emphasis of the manuscript is the role of noise on the MFPT. A non-monotonic dependence of the MPFT on the noise strength was shown in W. Braun, P. C. Matthews, R. Thul, First-passage times in integrate-and-fire neurons with stochastic thresholds, Physical Review E, 91, 21953 (2015), where the MFPT exceeds the deterministic value for small noise strength and then becomes smaller for larger noise strength. The paper also contains analytical results for computing the full FPT distribution.*
- ▶ We thank the reviewer for bringing our attention to the work of W. Braun, P. C. Matthews, and R. Thul regarding the non-monotonic dependence of the mean first-passage time (MFPT) on noise strength. This is indeed a valuable reference, and provides additional insight into the impact of noise on first-passage times. We have now incorporated this reference into the conclusion of the revised manuscript (See Page 11 and Ref 84 in the main text).
- (4) *In Figure 1, the y-axis of the FPT distribution is slightly misleading. I recommend to move the origin of the coordinate system for the FPT distribution to align with the coordinate system of the time course. In this way, the full FPT distribution can be shown, highlighting that it is zero at $t=0$ and positively skewed.*
- ▶ Our original intention for the figure as a schematic only was not clearly conveyed, and we acknowledge that the y-axis could be interpreted as misleading in its current form. To address this, we have revised Figure 1(B) to better align the FPT distribution with the trajectories, providing a more accurate representation. Specifically, the FPT is now zero at $t = 0$, and the shape of the distribution is positively skewed. We thank the reviewers for their suggestion.
- (5) *On p3, the authors write “we are interested in the distribution of first-passage times”. Since the paper focusses on MFPT and Section 2.1 only considers moments, I suggest to reword the above sentence as e.g. “we are interested in first-passage times”*
- ▶ We agree, and have amended this sentence to “we are interested in first-passage times”.
- (6) *When introducing Y , it might be worth stating that the state-space is discrete (as the entire derivation could be done for a continuous state space as well, but the derivation is only written for a discrete state space). After Equation (1), I would say $z \notin Y$ for completeness. I would like to add that I enjoyed the new proof for the moments of the FPT distribution.*
- ▶ We have now specified that the state space \mathbf{Y} is discrete, and following Equation (1) we have added that $\mathbf{z} \notin \mathbf{Y}$, as suggested. We are delighted that the new proof was appreciated!

REFERENCES

- [1] A Goldbeter. A model for circadian oscillations in the drosophila period protein (PER). *Proc. Biol. Sci.*, 261(1362):319–324, September 1995.
- [2] J C Leloup and A Goldbeter. Modeling the molecular regulatory mechanism of circadian rhythms in drosophila. *Bioessays*, 22(1):84–93, January 2000.
- [3] Peter Ruoff, Jennifer J Loros, and Jay C Dunlap. The relationship between FRQ-protein stability and temperature compensation in the neurospora circadian clock. *Proc. Natl. Acad. Sci. U. S. A.*, 102(49):17681–17686, December 2005.
- [4] Amit Singh, Congxin Li, Axel C R Diernfellner, Thomas Höfer, and Michael Brunner. Data-driven modelling captures dynamics of the circadian clock of neurospora crassa. *PLoS Comput. Biol.*, 18(8):e1010331, August 2022.
- [5] Olivier Pourquié. The segmentation clock: Converting embryonic time into spatial pattern. *Science*, 301(5631):328–330, 2003.
- [6] Kuan-Wei Chen, Kang-Ling Liao, and Chih-Wen Shih. The kinetics in mathematical models on segmentation clock genes in zebrafish. *Journal of Mathematical Biology*, 76:97–150, 2018.
- [7] Hiromi Hirata, Yasumasa Bessho, Hiroshi Kokubu, Yoshito Masamizu, Shuichi Yamada, Julian Lewis, and Ryoichiro Kageyama. Instability of hes7 protein is crucial for the somite segmentation clock. *Nat. Genet.*, 36(7):750–754, May 2004.
- [8] O Pourquie. The vertebrate segmentation clock. *J. Anat.*, 199(Pt 1-2):169–175, 2001.
- [9] Julian Lewis. Autoinhibition with transcriptional delay: A simple mechanism for the zebrafish somitogenesis oscillator. *Curr. Biol.*, 13(16):1398–1408, August 2003.

- [10] Daniel Forger, Didier Gonze, David Virshup, and David K Welsh. Beyond intuitive modeling: combining biophysical models with innovative experiments to move the circadian clock field forward. *J. Biol. Rhythms*, 22(3):200–210, June 2007.
- [11] Daniel T Gillespie. Deterministic limit of stochastic chemical kinetics. *The Journal of Physical Chemistry B*, 113(6):1640–1644, 2009.

Email address: lucy.ham@unimelb.edu.au

Email address: mcoomer@student.unimelb.edu.au

Email address: kaan.ocal@ed.ac.uk

Email address: ramon.grima@ed.ac.uk

Email address: mstumpf@unimelb.edu.au

REVIEWERS' COMMENTS

Reviewer #1 (Remarks to the Author):

The authors have adressed all my comments.

Reviewer #2 (Remarks to the Author):

The authors have addressed my concerns. This is a high quality paper deserving of publication.

Reviewer #3 (Remarks to the Author):

Reviewer #4 (Remarks to the Author):

The authors addressed all my comments satisfactorily. I am happy for the manuscript to. be published.